# Task-Agnostic Undesirable Feature Deactivation Using Out-of-Distribution Data

**Dongmin Park**[1], **Hwanjun Song**[2], **MinSeok Kim**[1], **Jae-Gil Lee**[1]*
[1] KAIST, [2] NAVER AI Lab
Republic of Korea
{dongminpark, minseokkim, jaegil}@kaist.ac.kr, hwanjun.song@navercorp.com

## Abstract

A deep neural network (DNN) has achieved great success in many machine learning tasks by virtue of its high expressive power. However, its prediction can be easily biased to undesirable features, which are not essential for solving the target task and are even imperceptible to a human, thereby resulting in poor generalization. Leveraging plenty of undesirable features in *out-of-distribution (OOD)* examples has emerged as a potential solution for de-biasing such features, and a recent study shows that softmax-level calibration of OOD examples can successfully remove the contribution of undesirable features to the last fully-connected layer of a classifier. However, its applicability is confined to the classification task, and its impact on a DNN feature extractor is not properly investigated. In this paper, we propose TAUFE, a novel regularizer that deactivates many undesirable features using OOD examples in the *feature extraction* layer and thus removes the dependency on the task-specific softmax layer. To show the task-agnostic nature of TAUFE, we rigorously validate its performance on three tasks, classification, regression, and a mix of them, on CIFAR-10, CIFAR-100, ImageNet, CUB200, and CAR datasets. The results demonstrate that TAUFE consistently outperforms the state-of-the-art method as well as the baselines without regularization.

## 1 Introduction

*Undesirable features*, which are informally defined as those not relevant to a target task, frequently appear in training data; for example, the background is an undesirable feature for classifying animals in images. The undesirable features are mainly caused by the statistical bias in *in-distribution* training data. In fact, many undesirable features are statistically correlated with labels, even though they are unnecessary and sometimes even harmful for the target task [1]; for example, the "desert" background feature is correlated with "camels" because the camels frequently appear in a desert. However, such undesirable features (e.g., desert background) rather yield unreliable predictions because they are easily shifted in other unseen data (e.g., images of the camels on the road).

Meanwhile, deep neural networks (DNNs) are known to overly capture any high-frequency data components which are even imperceptible to a human [2, 3]. This property is attributed to the vulnerability of DNNs that can totally overfit to random labels or adversarial examples owing to their extremely high capacity [3, 4, 5, 6]. Accordingly, DNNs are easily biased toward the *undesirable* features as well, thereby often showing unsatisfactory generalization to unseen examples [7]. Thus, it is very important to prevent overfitting to the undesirable features.

In this regard, a few research efforts have been devoted to remove the negative influence of undesirable features by leveraging *out-of-distribution (OOD)* data [8, 9]. Under the assumption that in-distribution and OOD data *share* undesirable features, OOD data is treated as a useful resource to alleviate the aforementioned undesirable bias. Notably, a recent study [8] proposed a *softmax-level* calibration,

---

*Corresponding author.

35th Conference on Neural Information Processing Systems (NeurIPS 2021).

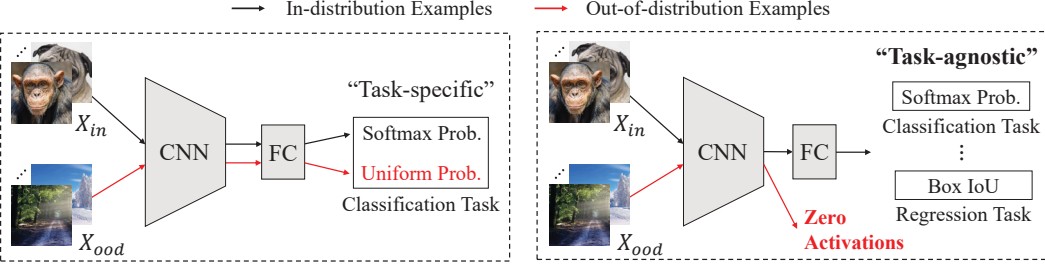

(a) Softmax-level calibration.      (b) Feature-level calibration (TAUFE).

Figure 1: Comparison of softmax-level and feature-level calibrations.

which assigns uniform softmax probabilities to all possible labels for all examples in OOD data. Although this approach shows decent de-biasing performance in the classification task, the softmax-level calibration has *two* limitations:

- **Lack of Flexibility:** The softmax-level calibration is designated only for the classification task. However, the bias toward undesirable features occurs in numerous tasks, such as object localization and bounding box regression. Therefore, a flexible, task-agnostic approach is required for easily supporting other downstream tasks too.

- **Feature Entanglement:** Even desirable features can be entangled with undesirable ones by assigning the uniform softmax probability invariably to all possible labels for OOD examples. Thus, the negative influence of the undesirable features is not perfectly removed because they still remain and affect the activation of desirable features (See § 3.3 for details).

In this paper, we propose a novel *task-agnostic* and *feature-level* calibration method, called TAUFE (Task-Agnostic Undesirable Feature dEactivation), which explicitly forces a model to produce *zero* values for many undesirable features in OOD examples. Differently from the softmax-level calibration that regularizes the *classification* layer (Figure 1(a)), TAUFE exploits the *penultimate* layer right before the classification layer and deactivates its activation only for OOD examples (Figure 1(b)). Thus, TAUFE is applicable to any task that requires another task-specific layer other than the classification layer, and the undesirable features are removed in the feature level without feature entanglement. The superiority of the proposed *feature-level* calibration over the *softmax-level* calibration is proven by theoretical and empirical analysis of the feature activation of the penultimate layer.

To validate its general efficacy, we conducted extensive experiments through *three* tasks: (i) image classification for classification; (ii) bounding box regression for regression; and (iii) weakly supervised object localization (WSOL) for a mix of them. We tested multiple pairs of in-distribution and OOD data: CIFAR-10, CIFAR-100, ImageNet, CUB200, and CAR for in-distribution; and SVHN, LSUN, and Places365 for OOD. The experiment results demonstrate that TAUFE consistently outperforms the softmax-level calibrator [8] by up to $9.88\%$ for classification and by up to $8.03\%$ for the mix of classification and regression.

Our main contributions are summarized as follows:

1. We propose a simple yet effective method, TAUFE, to deactivate undesirable features in learning, which is easily applicable to any standard learning task with recent DNNs.

2. We provide an insight on how feature-level and softmax-level calibration differently affect feature extraction by theoretic and empirical analysis on the penultimate layer activation.

3. We validate the task-agnostic nature of TAUFE through three tasks and show its performance advantage over the state-of-the-art method.

## 2 Background and Related Work

**Negative Impact of Undesirable Features.**      DNNs tend to overly capture all available signals from training data even when they are not essential for solving a given task [2, 3]. The occurrence of the undesirable features and their negative impact have been recently witnessed in various types of learning tasks. In image classification, a classification model often uses background or texture features as an undesirable shortcut for making a prediction instead of using the intrinsic shape of a target class [3, 7]. In object detection, a detector model easily overfits to the background features for

localizing target objects in a scene [10, 11]. In video action recognition, a recognition model often relies on static cues in a single frame rather than temporal actions over consecutive frames [12, 13]. In natural language processing (NLP) tasks, a language model often makes its predictions based on frequent but meaningless words instead of using semantically meaningful words [14].

**Connection with Adversarial Examples.** DNNs are easily deceived by adversarial perturbations of the inputs, so-called adversarial examples [15]. Differently from standard learning, the undesirable features are maliciously added and then make the model incur more errors. In addition, it is widely known that such adversarial perturbations are transferable even from different domains [16]; that is, an adversarial attack can drastically degrade the generalization capability of the classifier without knowing its internals [16]. To remedy this problem, the use of OOD examples has gained great attention in that they enhance the robustness against the adversarial examples by preventing the model from overfitting to the undesirable features [8].

**Removing Undesirable Feature Contribution.** Numerous studies have attempted to prevent the overfitting to the undesirable features in standard supervised learning tasks. A typical way is *de-biasing*, which removes the undesirable feature contribution based on the pre-defined bias for the target task. Geirhos et al. [7] took advantage of data augmentation techniques to generate de-biased examples from training data. Lee et al. [17] and Shetty et al. [18] synthesized de-biased examples by leveraging a generative model for image stylization or object removal. Wang et al. [19] quantified the local feature bias by using the neural gray-level co-occurrence matrix. Bahng et al. [1] proposed a framework that leverages a bias-characterizing model to remove pixel-level local undesirable features. This family of methods successfully removes the *pre-defined* bias from the undesirable features, but is not generalizable to other types of bias. Even worse, it is hard to identify the types of undesirable features in advance since they are not comprehensible even to a human.

In this regard, motivated by the transferability of undesirable features in different domains, the usefulness of OOD examples for de-biasing was started to be discussed. Although the representative softmax calibrator, OAT [8], does not need a pre-defined bias type, it suffers from the two limitations, lack of flexibility and feature entanglement. Many aspects, such as high generalizability and in-depth theoretical analysis, are yet to be explored.

# 3 Proposed Method: TAUFE

In this section, we first formulate the problem following the setup in the relevant literature [3, 8, 9] and then describe our method TAUFE. Moreover, we provide a theoretic analysis with empirical evidence on how the softmax-level and feature-level calibrations work differently at the penultimate layer in the perspective of feature extraction.

## 3.1 Problem Formulation

Let $\mathcal{D} = \{x_i, y_i\}_{i=1}^N$ be the target data obtained from a joint distribution over $\mathcal{X} \times \mathcal{Y}$, where $\mathcal{X}$ is the in-distribution example space and $\mathcal{Y}$ is the target label space. A DNN model consists of a general feature extractor $f_\phi : \mathcal{X} \to \mathcal{Z} \in \mathbb{R}^d$ and a task-specific layer $g_\theta : \mathcal{Z} \to \mathcal{Y}$. Then, the feature extractor is considered as a compound of $d$ sub-feature extractors $f_{\phi_j}$ such that $f_\phi(x) = \{f_{\phi_1}(x), \ldots, f_{\phi_d}(x)\}$ where $f_{\phi_j} : \mathcal{X} \to \mathbb{R}$. A *feature* is defined to be a function mapping from the example space $\mathcal{X}$ to a real number, and a set of the features is denoted by $\mathcal{F} = \{f \in f_\phi : \mathcal{X} \to \mathbb{R}\}$.

We now formalize the desirableness of a feature. Let $\tilde{\mathcal{D}} = \{\tilde{x}_i\}_{i=1}^M$ be the *out-of-distribution (OOD)* data obtained from a distribution over the OOD example space $\tilde{\mathcal{X}}$. Then, *undesirable* and *desirable* features are defined by Definitions 3.1 and 3.2, respectively.

*Definition 3.1* (UNDESIRABLE FEATURE). For each example $\tilde{x}$ in the OOD data $\tilde{\mathcal{D}}$, we call a feature *undesirable* if it is highly correlated with at least one true label in expectation. Thus, the set $\mathcal{F}_{undesirable}(\rho)$ of undesirable features is defined by

$$\mathcal{F}_{undesirable}(\rho) = \Big\{ f \in \mathcal{F} : \mathbb{E}_{\tilde{x} \in \tilde{\mathcal{D}}}\big[ \max_{y \in \mathcal{Y}} |\mathrm{Corr}\big(f(\tilde{x}), y\big)| \big] \geq \rho \Big\}, \tag{1}$$

where Corr is a function to produce the correlation between two given inputs (e.g., $R^2$) and $\rho$ is a constant threshold. $| \cdot |$ is an absolute value function to convert a negative correlation into a positive one. Intuitively speaking, an undesirable feature influences the model's decision-making even if it is not relevant to the target task (i.e., OOD examples).

*Definition 3.2* (DESIRABLE FEATURE). For each example $x$ and its corresponding label $y$ in the in-distribution data $\mathcal{D}$, we call a feature *desirable* if it is highly correlated with the true label in expectation and does not belong to $\mathcal{F}_{undesirable}(\rho)$. Thus, the set $\mathcal{F}_{desirable}(\epsilon)$ of desirable features is defined by

$$\mathcal{F}_{desirable}(\epsilon) = \left\{ f \in \mathcal{F}/\mathcal{F}_{undesirable}(\rho) : \mathbb{E}_{(x,y)\in\mathcal{D}}\big[\, |\mathrm{Corr}\big(f(x),y\big)|\,\big] \geq \epsilon \right\}, \qquad (2)$$

where $\epsilon$ is a constant threshold; Corr and $\rho$ are the same as those for Definition 3.1.

Note that Definitions 3.1 and 3.2 are generally applicable to any supervised learning tasks including classification and regression. By these definitions, a feature vector obtained by the feature extractor could be a mixture of desirable and undesirable features. DNNs can totally memorize even undesirable features owing to their high expressive power, leading to the statistical bias in in-distribution training data. Therefore, the main challenge is to prevent the problem of biasing toward the undesirable features, which will be discussed in the next section.

## 3.2 Main Concept: Feature-Level Calibration

We introduce the notion of the *feature-level* calibration, which directly manipulates the activations of the general feature extractor $f_\phi$. The key idea is to force the feature activations of all OOD examples to be zero vectors, thereby preventing the undesirable features from being carried over into the last task-specific layer $g_\theta$. Equation (3) shows the difference in the objective function among standard learning, OAT (softmax-level calibration) [8], and TAUFE (feature-level calibration):

$$\text{STANDARD: } \min_{\phi,\theta} \ \mathbb{E}_{(x,y)\in\mathcal{D}} \left[ \ell\Big(g_\theta\big(f_\phi(x)\big), y\Big) \right],$$

$$\text{OAT: } \min_{\phi,\theta} \ \mathbb{E}_{(x,y)\in\mathcal{D}} \left[ \ell\Big(g_\theta\big(f_\phi(x)\big), y\Big) \right] + \lambda \ \mathbb{E}_{\tilde{x}\in\tilde{\mathcal{D}}} \left[ \ell\Big(g_\theta\big(f_\phi(x)\big), t_{\text{unif}}\Big) \right], \qquad (3)$$

$$\text{TAUFE: } \min_{\phi,\theta} \ \mathbb{E}_{(x,y)\in\mathcal{D}} \left[ \ell\Big(g_\theta\big(f_\phi(x)\big), y\Big) \right] + \lambda \ \mathbb{E}_{\tilde{x}\in\tilde{\mathcal{D}}} \left[ ||f_\phi(\tilde{x})||_2^2 \right],$$

where $t_{\text{unif}} = [1/c, \dots, 1/c]$ and $\ell$ is the target loss for each original task (e.g., cross-entropy loss for classification or mean squared error (MSE) loss for regression). The first term is the same for all three methods, but there is a difference in the second term. Both OAT and TAUFE use the OOD examples (i.e., $\tilde{\mathcal{D}}$) to avoid the memorization of the undesirable features, but only TAUFE is not dependent on the task-specific layer $g_\theta$ in its regularization mechanism. Therefore, this feature-level calibration is easily applicable to any type of tasks for practical use and, at the same time, reduces the impact of the undesirable features on the model's prediction.

More importantly, TAUFE is remarkably simple. We contend that its simplicity should be a strong benefit because simple regularization often makes a huge impact and gains widespread use, as witnessed by weight decay and batch normalization.

Algorithm 1 describes the overall procedure of TAUFE, which is self-explanatory.

---

**Algorithm 1** TAUFE

---

INPUT: $\mathcal{D}$: target data, $\tilde{\mathcal{D}}$: OOD data, $epochs$: total number of epochs, $b$: batch size
OUTPUT: $\phi_t, \theta_t$: network parameters
1: $t \leftarrow 1; \phi_t, \theta_t \leftarrow$ Initialize the network parameters;
2: **for** $i = 1$ **to** $N$ **do**
3:     **for** $j = 1$ **to** $|\mathcal{D}|/b$ **do**
4:         Draw a mini-batch $\mathcal{B}$ from $\mathcal{D}$; /* A target mini-batch. */
5:         Draw a mini-batch $\tilde{\mathcal{B}}$ from $\tilde{\mathcal{D}}$; /* An OOD mini-batch. */
6:         /* Update for the feature extractor by the feature-level calibration. */
7:         $\phi_{t+1} = \phi_t - \alpha\nabla_\phi\big(\mathbb{E}_{(x,y)\in\mathcal{B}}\left[\ell\big(g_{\theta_t}\big(f_{\phi_t}(x)\big), y\big)\right] + \lambda\,\mathbb{E}_{\tilde{x}\in\tilde{\mathcal{B}}}\left[||f_{\phi_t}(\tilde{x})||_2^2\right]\big)$
8:         /* Update for the task-specific model by the standard manner. */
9:         $\theta_{t+1} = \theta_t - \alpha\nabla_\theta\,\mathbb{E}_{(x,y)\in\mathcal{B}}\left[\ell\big(g_{\theta_t}\big(f_{\phi_t}(x)\big), y\big)\right]$;
10:        $t \leftarrow t + 1$;
11: **return** $\phi_t, \theta_t$

---

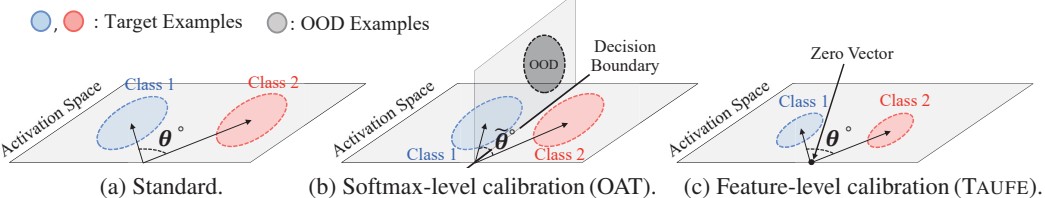

Figure 2: Effect of the softmax-level and feature-level calibrations on the penultimate layer activations.

### 3.3 Theoretical and Empirical Analysis

We analyze that the feature-level calibration works better than the softmax-level calibration in terms of feature disentanglement on the penultimate layer activations. The use of OOD examples with the softmax-level calibration has been theoretically proven to remove undesirable feature contributions to the last linear classifier [8]. However, the proof holds under the strong assumption that desirable and undesirable features should be disentangled perfectly before entering the last classifier layer. Because this assumption does not hold in practice, we provide an in-depth analysis on the use of OOD examples on the perspective of feature extraction without any assumption.

**Theoretic Analysis of Softmax-Level Calibration.** The effect of the softmax-level calibration is tightly related with label smoothing, which is a regularization technique [20] that uses the target label combined with a uniform mixture over all possible labels. Let $z$ be the penultimate layer activation and $w_k$ be a weight row-vector of the last linear classifier assigned to the $k$-th class. Then, the logit $z^T w_k$ for the $k$-th class can be thought of the negative *Euclidean distance* between $z$ and a weight template $w_k$, because $||z - w_k||^2 = z^T z + w_k^T w_k - 2z^T w_k$ where $z^T z$ and $w_k^T w_k$ are usually constant across classes. Therefore, when OAT assigns the uniform softmax probability to OOD examples, each logit $z^T w_k$ is forced into being the same value, which means that the penultimate layer activation $z$ is *equally distant* to the templates (i.e., clusters) of all classes.

As shown in Figure 2(b), forcing all OOD examples into being equally distant to all class templates is mathematically equivalent to locating them on the hyper-plane across the decision boundaries. While the hyper-plane is orthogonal to the space composed of desirable features, it is likely onto a decision boundary. Accordingly, the undesirable features move the activations of the desirable features toward a decision boundary, and the two types of features are entangled. Overall, although the softmax-level calibration helps remove undesirable features, it partially entangles the undesirable features with the desirable features, degrading the prediction performance.

**Theoretic Analysis of Feature-Level Calibration.** In contrast to the softmax-level calibration, the feature-level calibration explicitly forces the activations of all OOD examples into approaching the *zero* vector [21], as shown in Figure 2(c). This regularization reduces the norm of all target examples without changing the angle between the activations for different classes if they share undesirable features. Since this angle plays a decisive role for classification [22], the feature-level calibration removes the undesirable features while effectively maintaining the disentanglement between desirable and undesirable features. See §B of the supplementary material for in-depth theoretical analysis.

**Empirical Analysis.** To empirically support our analysis, in Table 1, we quantitatively calculate the cosine similarity of activations across all in-distribution classes. Compared with the standard learning method, TAUFE (feature-level calibration) decreases the cosine similarity between classes, whereas OAT (softmax-level calibration) rather increases the cosine similarity. That is, OAT is prone to move the activations

Table 1: Average cosine similarity between all activation pairs across different classes on CIFAR-10 for Standard, OAT, and TAUFE.

| Datasets | | Methods | | |
|---|---|---|---|---|
| In-dist. | Out-of-dist. | Standard | OAT | TAUFE |
| CIFAR-10 | LSUN | 0.116 | 0.286 | 0.095 |

of in-distribution examples toward the decision boundary, though it reduces the negative effect of the undesirable features on the classification task. In contrast, it is noteworthy that TAUFE renders the activations of different in-distribution classes more distinguishable. Furthermore, we visualize the penultimate activations of the two in-distribution classes in CIFAR-10 together with those of OOD examples in Figure 3. As shown in Figure 3(b), OAT simply locates the activations of OOD examples around the decision boundary. However, TAUFE forces them into the zero vector without

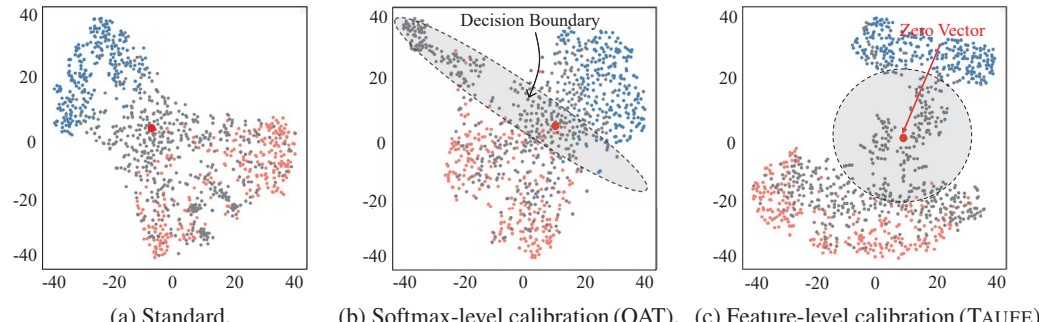

| (a) Standard. | (b) Softmax-level calibration (OAT). | (c) Feature-level calibration (TAUFE). |

Figure 3: TSNE visualization of the penultimate layer activations. In-distribution examples are in pink for the automobile class and in blue for the bird class, while all OOD examples are in grey.

much change in the angles between different classes. Therefore, the empirical evidences confirm that TAUFE successfully reduces the negative effect of undesirable features on the classification task.

## 4 Experiments

We compare TAUFE with the standard learning method (denoted as "Standard") and the state-of-the-art method OAT [8]. Standard trains the network without any calibration process for OOD examples. In addition, we include the few-shot learning settings because DNNs are easily biased toward the undesirable features especially when the number of training examples is small. All methods are implemented with PyTorch 1.8.0 and executed using four NVIDIA Tesla V100 GPUs. For reproducibility, we provide the source code at `https://github.com/kaist-dmlab/TAUFE`. In support of reliable evaluations, we repeat every test *five* times and report the average.

To show high flexibility to diverse types of tasks, we rigorously validate the efficacy of TAUFE for *three* popular visual recognition tasks: (i) image classification, (ii) bounding box regression, and (iii) weakly supervised object localization (WSOL). Please note that OAT does not support the bounding box regression task because of the absence of the softmax layer.

### 4.1 Task I: Image Classification

**Dataset.** We choose CIFAR-10, CIFAR-100 [23], and ImageNet [24] for the target in-distribution data. For the CIFAR datasets, two out-of-distribution datasets are carefully mixed for evaluation—LSUN [25], a scene understanding dataset of 59M images with 10 classes such as bedroom and living room, and SVHN [26], a real-world house numbers dataset of 70K images with 10 classes. The ImageNet dataset is divided into 12K images of 10 randomly selected classes (ImageNet-10) and 1.1M images of the rest 990 classes (ImageNet-990); the former and the latter are used as in-distribution data and OOD data, respectively. A large-scale collection of place scene images with 365 classes, Places365 [27], is also used as another OOD data for ImageNet-10.

**Training Configuration.** For CIFAR datasets, ResNet-18 [28] is trained from scratch for 200 epochs using SGD with a momentum of 0.9, a batch size of 64, a weight decay of 0.0005. To support the original resolution, we drop the first pooling layer and change the first convolution layer with a kernel of 3, a stride size of 1, and a padding size of 1. An initial learning rate of 0.1 is decayed by a factor of 10 at 100-th and 150-th epochs, following the same configuration in OAT [8]. For the ImageNet-10 dataset, ResNet-50 is used without any modification, but the resolution of ImageNet-10 is resized into 64×64 and 224×224 in order to see the effect of different resolutions. Resized random crops and random horizontal flips are applied for data augmentation.

TAUFE requires only one additional hyperparameter, the scaling factor $\lambda$ for the feature-level calibration in Equation (3). The value of $\lambda$ is set to be 0.1 and 0.01 for CIFARs and ImageNet-10, respectively, where the best values are obtained via a grid search. The corresponding hyperparameter in OAT for softmax-level calibration is set to be 1, following the original paper. In addition, both few-shot and full-shot learning settings are considered for evaluation. Given the number $N$ of the examples for use in few-shot learning, $N$ examples are randomly sampled over all classes from both in-distribution and OOD data, and thus $2N$ examples in total are used for training. For full-shot learning, $N$ is set to be the total number of training examples in the target in-distribution data.

Table 2: Classification accuracy (%) of TAUFE compared with Standard and OAT on two CIFARs (32×32), ImageNet-10 (64×64), and ImageNet-10 (224×224) under few-shot and full-shot learning settings. The highest values are marked in bold.

| Datasets | | Methods | # Examples ($N$) | | | | |
|---|---|---|---|---|---|---|---|
| In-dist. | Out-of-dist. | | 500 | 1,000 | 2,500 | 5,000 | Full-shot |
| CIFAR-10 (32×32) | – | Standard | 38.58 | 52.63 | 72.94 | 82.38 | 94.22 |
| | SVHN | OAT | 40.55 | 52.80 | 73.24 | 82.56 | 94.38 |
| | | TAUFE | 41.58 | 56.72 | 73.61 | 82.88 | 94.45 |
| | LSUN | OAT | 40.73 | 53.16 | 73.51 | 82.71 | 94.61 |
| | | TAUFE | **42.51** | **56.79** | **74.15** | **83.73** | **95.02** |
| CIFAR-100 (32×32) | – | Standard | 11.07 | 13.99 | 24.28 | 41.47 | 73.84 |
| | SVHN | OAT | 10.92 | 14.56 | 24.67 | 42.21 | 74.82 |
| | | TAUFE | 11.30 | 15.13 | 24.91 | 43.61 | 75.38 |
| | LSUN | OAT | 11.27 | 15.24 | 24.75 | 43.09 | 75.15 |
| | | TAUFE | **12.26** | **15.97** | **25.36** | **44.50** | **75.69** |
| ImageNet-10 (64×64) | – | Standard | 38.82 | 43.66 | 56.17 | 66.80 | 78.30 |
| | ImageNet-990 | OAT | 38.95 | 44.09 | 57.29 | 69.41 | 79.29 |
| | | TAUFE | 42.80 | 46.04 | **60.40** | **70.51** | **81.09** |
| | Places365 | OAT | 41.06 | 43.81 | 56.47 | 67.20 | 79.30 |
| | | TAUFE | **43.25** | **47.61** | 60.02 | 68.25 | 80.89 |
| ImageNet-10 (224×224) | – | Standard | 44.82 | 56.29 | 73.60 | 82.49 | 86.97 |
| | ImageNet-990 | OAT | 46.41 | 58.66 | 75.62 | 83.6 | 87.66 |
| | | TAUFE | 48.39 | 59.06 | 76.47 | **85.05** | **89.24** |
| | Places365 | OAT | 48.10 | 56.88 | 74.98 | 83.41 | 88.78 |
| | | TAUFE | **50.08** | **59.27** | **77.22** | 84.81 | 89.06 |

**Performance Comparison.** Table 2 shows the classification accuracy of the three methods under few-shot and full-shot learning settings. Overall, TAUFE shows the highest classification accuracy at any few-shot settings for all datasets. Specifically, TAUFE outperforms OAT by $0.07\%$ to $9.88\%$, though OAT also shows consistent performance improvement. OAT's lower performance is attributed to the property that it is prone to force the activations of in-distribution examples toward the decision boundary as analyzed in § 3.3. Adding LSUN as OOD for CIFARs is more effective than adding SVHN, because LSUN is more similar to CIFARs than SVHN, thus sharing more undesirable features. For ImageNet-10, adding Places365 is more effective than adding ImageNet-990 when the number of training examples is not enough, but adding ImageNet-990 becomes more effective as the size of training data increases. Because ImageNet-990 has more diverse background scenes than Places365, we conjecture that the effect of Places365 saturates faster than that of ImageNet-990 as more OOD examples are exposed to the DNN model. Besides, no significant difference is observed depending on the resolution of ImageNet-10.

**Performance with Semi-Supervised Learning.** We use a *semi-supervised learning* framework for a baseline in addition to the standard supervised learning framework, because TAUFE can also improve the accuracy of a semi-supervised classifier by deactivating the undesirable features. MixMatch [29], a popular semi-supervised learner for image classification, is enhanced with TAUFE by up to $2.02\%$ and $2.34\%$, respectively, on two CIFAR datasets. § 7.1 shows the details.

**Effect on Adversarial Robustness.** We investigate the effect of TAUFE on *adversarial robustness*. Overall, TAUFE improves the accuracy on adversarial examples by up to $2.76\%$ when adding the LSUN dataset as OOD. § 7.2 shows the details.

**Effect on OOD Detection.** As TAUFE is not intended to detect OOD examples, its effect on *OOD detection* is not noticeable, as shown in § C of the supplementary material.

### 4.2 Task II: Bounding Box Regression

Bounding box regression is an essential sub-task for object localization and object detection. We compare TAUFE with only Standard because OAT does not work for regression.

Table 3: IoU (%) of TAUFE compared with Standard on CUB200 (224×224) and CAR (224×224) under few-shot and full-shot learning settings. The highest values are marked in bold.

| Datasets | | Methods | L1 | | | L1-IoU | | | D-IoU | | |
|---|---|---|---|---|---|---|---|---|---|---|---|
| In-dist. | Out-of-dist. | | # Examples ($N$) | | | # Examples ($N$) | | | # Examples ($N$) | | |
| | | | 2,000 | 4,000 | Full | 2,000 | 4,000 | Full | 2,000 | 4,000 | Full |
| CUB200 (224×224) | – | Standard | 66.41 | 73.10 | 76.42 | 66.57 | 73.28 | 76.67 | 66.82 | 73.18 | 76.57 |
| | ImageNet | TAUFE | **67.16** | **74.31** | **77.12** | **67.22** | **74.40** | **77.24** | **67.03** | **74.22** | **77.00** |
| | Places365 | TAUFE | 66.70 | 73.55 | 76.86 | 66.87 | 73.63 | 77.01 | 66.88 | 73.66 | 76.88 |
| CAR (224×224) | – | Standard | 83.06 | 85.50 | 90.56 | 83.52 | 86.54 | 91.25 | 83.62 | 87.93 | 91.09 |
| | ImageNet | TAUFE | **85.23** | **87.82** | **91.32** | **85.82** | **89.11** | **91.40** | **85.30** | **89.06** | **91.35** |
| | Places365 | TAUFE | 84.26 | 87.59 | 90.86 | 84.73 | 88.83 | 91.28 | 84.60 | 88.64 | 91.20 |

Table 4: GT-known Loc of TAUFE compared with Standard on CUB200 (224×224) and CAR (224×224) under few-shot and full-shot learning settings. The highest values are marked in bold.

| Datasets | | Methods | # Examples ($N$) | | |
|---|---|---|---|---|---|
| In-dist. | Out-of-dist. | | 2,000 | 4,000 | Full-shot |
| CUB200 (224×224) | – | Standard | 54.45 | 58.37 | 64.02 |
| | ImageNet | OAT | 55.24 | 60.24 | 64.91 |
| | | TAUFE | **59.68** | **61.88** | **65.56** |
| | Places365 | OAT | 56.97 | 60.01 | 64.27 |
| | | TAUFE | 58.24 | 60.90 | 64.84 |
| CAR (32×32) | – | Standard | 62.09 | 67.12 | 70.54 |
| | ImageNet | OAT | 63.77 | 67.24 | 71.64 |
| | | TAUFE | **65.82** | **69.05** | **72.14** |
| | Places365 | OAT | 63.16 | 68.58 | 71.66 |
| | | TAUFE | 65.70 | 67.64 | 71.62 |

**Dataset.** Two datasets are used as the target in-distribution data for the bounding box regression task—Caltech-UCSD Birds-200-2011 (CUB200) [30], a collection of 6,033 bird images with 200 classes, and Standford Cars (Car) [31], a collection of 8,144 car images with 196 classes. For each image of 224×224 resolution, the two datasets contain a class label and bounding box coordinates of the top-left and bottom-right corners. ImageNet[2] and Places365 are used as OOD data.

**Training Configuration.** ResNet-50 is trained from scratch using SGD for 100 epochs. Following the prior work [32], the last classification layer in ResNet-50 is converted to a box regressor that predicts the bounding box coordinates of the top-left and bottom-right corners. In addition, we use three types of different loss functions: (i) L1, a L1-smooth loss, (ii) L1-IoU, a combination of L1 and IoU, and (iii) D-IoU [33], a combination of L1, IoU, and the normalized distance between the predicted box and the target box. The remaining configurations are the same as those in § 4.1.

**Evaluation Metric.** We adopt the Intersection over Union (IoU), which is the most widely-used metric for bounding box regression and defined by $\text{IoU}(b_i, \tilde{b}_i) = \frac{1}{k} \sum_{i=1}^{N} |b_i \cap \tilde{b}_i| / |b_i \cup \tilde{b}_i|$ where $b_i$ and $\tilde{b}_i$ are the ground-truth and predicted bounding boxes of the object in the $i$-th example.

**Performance Comparison.** Table 3 shows the IoU accuracy of the two methods. Overall, TAUFE consistently boosts the performance on bounding box regression for all datasets regardless of the loss type. Quantitatively, the box regression performance considerably improves with TAUFE by up to 2.97% when using L1-IoU. This result indicates that the use of OOD examples with the feature-level calibration indeed alleviates the undesirable bias problem. Interestingly, adding ImageNet as OOD for both CUB200 and CAR is more effective than adding Places365, possibly because ImageNet contains a higher number of classes which reflect more diverse undesirable features.

---

[2]All bird and vehicle relevant classes are excluded from the ImageNet dataset.

### 4.3 Task III: Weakly Supervised Object Localization (WSOL)

WSOL is a problem of localizing a salient foreground object in an image by using only weak supervision (i.e., image-level class labels). It can be considered as a mix of classification and regression because it uses class labels but aims at bounding box regression. The seminal WSOL work, class activation mapping (CAM) [34], has shown that the intermediate classifier activations focus on the most discriminative parts of the target object in the image. Thus, by simply averaging all local activations, we can estimate how much the corresponding pixels contribute to discriminate the object in the scene. CAM is used as the standard learning method. We refer the reader to the surveys [35] for more details about WSOL.

**Dataset.** Like the bounding box regression task, CUB200 and CAR are used as in-ditribution data, while Places365 and ImageNet are used as OOD data.

**Training Configuration.** ResNet-50 is trained from scratch for 100 epochs using SGD with a batch size of 64. An initial learning rate of 0.1 is decayed by a factor of 10 at 50-th and 75-th epochs. The remaining configurations are the same as those in § 4.1.

**Evaluation Metric.** We adopt the localization accuracy with known ground-truth class (GT-known Loc), which is the most widely-used metric for WSOL and defined by $\text{GT\_known\_Loc}(b_i, \tilde{b}_i) = \frac{1}{N} \sum_{i=1}^{N} \mathbb{1}\left(\text{IoU}(b_i, \tilde{b}_i) \geq \delta\right)$ where $b_i$ is the ground-truth box of the object in the $i$-th example and $\tilde{b}_i$ is the tightest box around the largest connected component of the activation mask for the $i$-th example. The IoU threshold $\delta$ is set to be 0.5, following the prior work [34, 35].

**Performance Comparison.** Table 4 shows the GT-known accuracy of the three methods. Overall, TAUFE shows the best localization accuracy at any few-shot settings for all datasets. Specifically, TAUFE outperforms OAT by 0.71% to 8.03%, though OAT also shows consistent performance improvement. This result indicates that TAUFE successfully removes the undesirable features such as background to locate an object in an image. Adding ImageNet as OOD is more effective than adding Places365 for the same reason. Besides, the performance gain of TAUFE over Standard is typically larger for few-shot learning than for full-shot learning, as observed in the other tasks.

## 5   Limitations and Future Work

Although TAUFE has shown consistent performance improvements in three types of real-world machine learning tasks, there are some issues that need to be further discussed. First, the effectiveness of an OOD dataset for given a target dataset and a task needs to be formulated theoretically. Owing to the transferability of undesirable features, any OOD dataset can be effective but its effectiveness varies as shown in § 4. The difference in the effectiveness may come from the amount of shared undesirable features between the target dataset and each OOD dataset. Therefore, formulating the effectiveness based on such factors is an interesting research direction. Second, the applicability of TAUFE need to be verified for a wide range of learning frameworks including self-supervised learning, semi-supervised learning, and meta-learning, because the bias toward undesirable features is likely to be observed regardless of the learning frameworks. Thus, we will clarify the outcome of TAUFE with varying the learning frameworks as future work.

## 6   Conclusion

In this paper, we propose TAUFE, a novel *task-agnostic* framework to reduce the bias toward undesirable features when training DNNs. Since the existing softmax-level calibration method confines its applicability to only the classification task, we overcome the limitation by introducing the *feature-level* calibration that directly manipulates the feature output of a general feature extractor (e.g., a convolutional neural network). To remove the effect of undesirable features on the final task-specific module, TAUFE simply deactivates many undesirable features extracted from the OOD data by regularizing them as zero vectors. Moreover, we provide an insight on how differently feature-level and softmax-level calibrations affect feature extraction by theoretic and empirical analysis on the penultimate layer activation. The consistent performance improvement on three types of tasks clearly demonstrates the task-agnostic nature of TAUFE. Overall, we believe that our work sheds the light on the usability of the OOD data in diverse tasks.

Table 5: Classification accuracy (%) of TAUFE under few-shot semi-supervised learning settings.

| In-dist. | CIFAR-10 | | | CIFAR-100 | | |
|---|---|---|---|---|---|---|
| Out-of-dist. | – | SVHN | LSUN | – | SVHN | LSUN |
| Methods | MixMatch | TAUFE$_{\text{Mix}}$ | TAUFE$_{\text{Mix}}$ | MixMatch | TAUFE$_{\text{Mix}}$ | TAUFE$_{\text{Mix}}$ |
| Accuracy | 88.32 | 90.02 | **90.10** | 51.38 | 52.32 | **52.58** |

Table 6: Accuracy (%) of TAUFE under the PGD adversarial attacker.

| In-dist. | CIFAR-10 | | | |
|---|---|---|---|---|
| Out-of-dist. | – | – | SVHN | LSUN |
| Methods | Standard | PGD | TAUFE$_{\text{PGD}}$ | TAUFE$_{\text{PGD}}$ |
| Clean. acc. / Adv. acc. | 94.22 / 0.00 | 71.22 / 44.26 | 72.35 / **44.37** | 72.31 / **45.48** |

# 7 Supplementary Experiments

## 7.1 Performance of TAUFE with Semi-Supervised Learning

**Baseline.** MixMatch [29] is one of the state-of-the-art semi-supervised learning frameworks for image classification. By leveraging unlabeled examples with automatic label guessing and mix-up techniques, MixMatch nearly reaches the accuracy of fully supervised learning using only a small number of labeled examples.

**Experiment Setting.** CIFAR-10 and CIFAR-100 are used for in-distribution datasets, and LSUN and SVHN are used for two OOD datasets. We use the default or best hyperparameter values suggested by the authors [29]. Specifically, the sharpening temperature $T$ is set to be 0.5, the number of augmentations $K$ to be 2, the Beta distribution parameter $\alpha$ to be 0.75, and the loss weight for unlabeled examples $\lambda_U$ to be 100. We fix the number of epochs to be 1,024 and the batch size to be 64, and linearly ramp up $\lambda_U$ in the first 16,000 optimization steps. We use 25 labeled examples per class as initially labeled data, because MixMatch was shown to nearly reach the full-supervision accuracy on that setting [29] .

**Result.** Table 5 shows the classification accuracy of TAUFE combined with MixMatch on two CIFAR datasets under few-shot settings—i.e., $N$=250 for CIFAR-10 and $N$=2,500 for CIFAR-100; TAUFE$_{\text{Mix}}$ represents the TAUFE combined with MixMatch. TAUFE$_{\text{Mix}}$ consistently improves the performance of MixMatch on two CIFAR datasets. Similar to the supervised learning in § 4.1, adding LSUN as OOD is more effective than adding SVHN; compared with MixMatch, the performance of TAUFE$_{\text{Mix}}$ is improved by up to 2.02% on CIFAR-10 and by up to 2.34% on CIFAR-100. This result shows that TAUFE successfully deactivates the negative effect of undesirable features even in the semi-supervised learning setting.

## 7.2 Effect of TAUFE on Adversarial Robustness

**Baseline.** We use the projected gradient descent (PGD) [36] attack / learning method, which employs an iterative procedure of the fast gradient sign method (FGSM) [37] to find the worst-case examples having the maximum training loss.

**Experiment Setting.** CIFAR-10 and CIFAR-100 are used for in-distribution datasets, and LSUN and SVHN are used for two OOD datasets which are exposed in the training phase. The hyperparameters of PGD are favorably set to be the best values reported in the original paper. The attack learning rate $\epsilon$ is set to be 2, and PGD$_n$ indicates the PGD attacks with $n$ iterative FGSM procedures. For adversarial learning, the adversarial examples generated by PGD$_7$ are used as the input of training. To measure the adversarial accuracy, the adversarial examples generated by PGD$_{100}$ are used for testing. This combination of step numbers was also used in the PGD work [36].

**Evaluation Metric.** The *clean accuracy* is the classification accuracy on the original test data, while the *adverserial accuracy* is that on the PGD$_{100}$ perturbed adversarial examples of the test data.

**Result.** Table 6 shows the adversarial robustness of TAUFE compared with the standard learning method. Overall, TAUFE improves the accuracy on adversarial examples by up to 2.76% when adding the LSUN dataset as OOD examples. This result indicates that the undesirable feature deactivation of TAUFE is helpful for the adversarial learning models.

## Acknowledgments and Disclosure of Funding

This work was supported by Institute of Information & Communications Technology Planning & Evaluation (IITP) grant funded by the Korea government (MSIT) (No. 2020-0-00862, DB4DL: High-Usability and Performance In-Memory Distributed DBMS for Deep Learning).

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
