# Task-Agnostic Undesirable Feature Deactivation
## Using Out-of-Distribution Data

### Supplementary Material

## A   Broader Impact

When labeling cost is high, DNNs are inevitably trained in few-shot learning settings. Then, the DNNs are prone to be biased toward undesirable features as shown in § 4. Thus, taking advantage of OOD examples for improving the generalization power is a very promising approach because no labeling is needed for the OOD examples. An application whose labeling cost is very high, e.g., bio-medical image analysis, is the sweet spot of our approach. Moreover, due to the task-agnostic nature of TAUFE, more tasks, not limited to classification, can benefit from the OOD examples. Meanwhile, we do not expect any potential negative societal impact of our work, because it is a regularization technique for improving the generalization power.

## B   In-Depth Theoretical Analysis

Because a DNN extracts any type of features if it is statistically correlated with the target label $y$, a feature vector $f_\phi$ of an in-distribution example $x$ contains both a desirable feature $f_{desirable}$ and an undesirable feature $f_{undesirable}$. That is, $f_\phi(x) = f_{desirable}(x) + f_{undesirable}(x)$, where $f \in \mathbb{R}^d$. On the other hand, because an OOD example $\tilde{x}$ does not contain any features that semantically indicate the target label $y$, $f_\phi(\tilde{x}) = f_{undesirable}(\tilde{x})$.

For ease of exposition, let's consider a binary classification setting. Let $x^+$ be an in-distribution example of the positive class and $x^-$ be an in-distribution example of the negative class. Then, via standard learning, $f_\phi(x^+) = f_{desirable}(x^+) + f_{undesirable}(x^+)$ and $f_\phi(x^-) = f_{desirable}(x^-) + f_{undesirable}(x^-)$. Because $f_{undesirable}(x^+)$ and $f_{undesirable}(x^-)$ are expected to share some features with $f_{undesirable}(\tilde{x})$, both OAT and TAUFE attempt to reduce their effect by the regularization on $f_\phi(\tilde{x}) = f_{undesirable}(\tilde{x})$. Here, $f_\phi(x^+)$ and $f_\phi(x^-)$ correspond to the red and blue circles, respectively, in Figure 2(a).

For notational simplicity, we denote $f_\phi(x^+)$ and $f_\phi(x^-)$ as follows:

$$\begin{aligned} f_\phi(x^+) &= f_{desirable}^+ + f_{undesirable}^+ \text{ and} \\ f_\phi(x^-) &= f_{desirable}^- + f_{undesirable}^-. \end{aligned} \tag{4}$$

**OAT.**  As analyzed in § 3.3, OAT regularizes the undesirable features from OOD examples being activated into the decision boundary. Thus, each class feature in Equation (4) is forced to be changed as follows:

$$\begin{aligned} f_\phi^{OAT}(x^+) &= f_{desirable}^+ + \left( \alpha \frac{(f_{desirable}^+ + f_{desirable}^-)}{2} + f_\perp^+ \right) \text{ and} \\ f_\phi^{OAT}(x^-) &= f_{desirable}^- + \left( \beta \frac{(f_{desirable}^+ + f_{desirable}^-)}{2} + f_\perp^- \right), \end{aligned} \tag{5}$$

where $\alpha, \beta \in \mathbb{R}$, $(f_{desirable}^+ + f_{desirable}^-)/2$ is a vector on the decision boundary, and $f_\perp$ is an orthogonal vector to the plane basis of $f_{desirable}^+$ and $f_{desirable}^-$. Then,

$$\begin{aligned} f_\phi^{OAT}(x^+) &= \left( 1 + \frac{\alpha}{2} \right) f_{desirable}^+ + \frac{\alpha}{2} f_{desirable}^- + f_\perp^+ \text{ and} \\ f_\phi^{OAT}(x^-) &= \frac{\beta}{2} f_{desirable}^+ + \left( 1 + \frac{\beta}{2} \right) f_{desirable}^- + f_\perp^-. \end{aligned} \tag{6}$$

Therefore, because the undesirable feature $f_{undesirable}$ moves the activation of the desirable feature toward the decision boundary, these two types of the features (i.e., $f_{desirable}$ and $f_{undesirable}$) tend to be entangled, as illustrated in Figure 2(b).

Table 7: OOD detection performance (%) of TAUFE compared with Standard using uncertainty-based and energy-based OOD detection methods.

| OOD detector | | Uncertainty | | | Energy | | |
|---|---|---|---|---|---|---|---|
| Dataset | Method | AUROC | AUPR$_{out}$ | FPR95 | AUROC | AUPR$_{out}$ | FPR95 |
| CIFAR-10 | Standard | 92.20 | 88.56 | 20.67 | 93.6 | 89.96 | 20.06 |
| | TAUFE | 92.17 | 89.71 | 22.96 | 93.37 | 90.08 | 22.88 |
| CIFAR-100 | Standard | 83.31 | 79.37 | 45.76 | 88.46 | 86.04 | 35.35 |
| | TAUFE | 82.03 | 79.21 | 47.89 | 88.42 | 88.56 | 37.82 |

**TAUFE.** As analyzed in § 3.3, TAUFE regularizes the undesirable features from OOD examples being deactivated on the feature space (i.e., toward the zero vector). Thus, each class feature in Equation (4) is forced to be changed as follows:

$$f_\phi^{\text{TAUFE}}(x^+) = f_{desirable}^+ + \vec{0} \text{ and}$$
$$f_\phi^{\text{TAUFE}}(x^-) = f_{desirable}^- + \vec{0}. \tag{7}$$

Therefore, this regularization does not affect the activation of $f_{desirable}$, as illustrated in Figure 2(c), thereby encouraging a prediction of a DNN to be solely based on the desirable features. This concludes the theoretical analysis of the novel L2 penalty term on TAUFE.

## C   Effect of TAUFE on OOD detection

We verify the effect of TAUFE on OOD detection, which aims at detecting out-of-distribution (OOD) examples in the test phase to support a trustworthy machine learning model.

**Baseline.** Numerous OOD detection methods have been proposed [38]. Here, we use two representative OOD detection methods—uncertainty-based [39] and energy-based [40]—to validate the effect of TAUFE on the OOD detection task.

**Experiment Setting.** CIFAR-10 and CIFAR-100 are used for in-distribution datasets; LSUN is used for exposing an OOD dataset in the training phase, and SVHN is used for measuring the detection performance in the test phase. The other training configurations are the same as those in §4.1.

**Evaluation Metric.** The OOD detection performance is commonly quantified using three metrics [39, 40]. *AUROC* is the area under the receiver operating characteristic, which is calculated by the area under the curve of the false positive rate (FPR) and the true positive rate (TPR). *AUPR$_{out}$* is the area under the curve of the precision and the recall, where they are calculated by considering OOD and in-distribution examples as positives and negatives, respectively. *FPR95* is the FPR at 95% of the TPR, which indicates the probability that an OOD example is misclassified as an in-distribution example when the TPR is 95%.

**Result.** Table 7 shows the OOD detection performance of the two representative OOD detection methods without and with TAUFE. According to the three metrics, the performance with TAUFE is slightly higher than or just comparable to that without TAUFE in both OOD detection methods. Overall, as TAUFE is not geared for OOD detection, it does not significantly affect the OOD detection performance on two CIFAR datasets.

## D   Details of Experiment Results

The source code for reproducing our experiment results is available at `https://github.com/kaist-dmlab/TAUFE`. In support of reliable validation, we show the standard deviation of our results for all three tasks in Tables 8, 9, and 10, respectively. These small deviations confirm that the results are stable with multiple (five) executions.

Table 8: Classification accuracy and standard deviation (%) of TAUFE on two CIFARs (32×32), ImageNet-10 (64×64), and ImageNet-10 (224×224).

| Datasets | | # Examples ($N$) | | | | |
|---|---|---|---|---|---|---|
| In-dist. | OOD | 500 | 1,000 | 2,500 | 5,000 | Full-shot |
| CIFAR-10 (32×32) | SVHN | 41.58±0.15 | 56.72±0.10 | 73.61±0.07 | 82.88±0.05 | 94.45±0.02 |
| | LSUN | **42.51**±0.11 | **56.79**±0.05 | **74.15**±0.04 | **83.73**±0.02 | **95.02**±0.03 |
| CIFAR-100 (32×32) | SVHN | 11.30±0.14 | 15.13±0.21 | 24.91±0.07 | 43.61±0.05 | 75.38±0.03 |
| | LSUN | **12.26**±0.09 | **15.97**±0.07 | **25.36**±0.04 | **44.50**±0.04 | **75.69**±0.02 |
| ImgNet-10 (64×64) | ImgNet-990 | 42.80±0.11 | 46.04±0.10 | **60.40**±0.07 | **70.51**±0.04 | **81.09**±0.03 |
| | Places365 | **43.25**±0.07 | **47.61**±0.08 | 60.02±0.07 | 68.25±0.06 | 80.89±0.04 |
| ImgNet-10 (224×224) | ImgNet-990 | 48.39±0.20 | 59.06±0.15 | 76.47±0.06 | **85.05**±0.03 | **89.24**±0.03 |
| | Places365 | **50.08**±0.16 | **59.27**±0.11 | **77.22**±0.03 | 84.81±0.04 | 89.06±0.02 |

Table 9: IoU and standard deviation (%) of TAUFE on CUB200 (224×224) and CAR (224×224) under few-shot and full-shot learning settings.

| Datasets | | L1 | | | L1-IoU | | |
|---|---|---|---|---|---|---|---|
| | | # Examples ($N$) | | | # Examples ($N$) | | |
| In-dist. | OOD | 2,000 | 4,000 | Full | 2,000 | 4,000 | Full |
| CUB200 (224×224) | ImageNet | **67.16**±0.05 | **74.31**±0.03 | **77.12**±0.02 | **67.22**±0.02 | **74.40**±0.04 | **77.24**±0.03 |
| | Places365 | 66.70±0.07 | 73.55±0.05 | 76.86±0.03 | 66.87±0.10 | 73.63±0.05 | 77.01±0.04 |
| CAR (224×224) | ImageNet | **85.23**±0.04 | **87.82**±0.02 | **91.32**±0.01 | **85.82**±0.05 | **89.11**±0.04 | **91.40**±0.03 |
| | Places365 | 84.26±0.06 | 87.59±0.07 | 90.86±0.02 | 84.73±0.05 | 88.83±0.04 | 91.28±0.02 |

Table 10: GT-known Loc and standard deviation of TAUFE on CUB200 (224×224) and CAR (224×224) under few-shot and full-shot learning settings.

| Datasets | | # Examples ($N$) | | |
|---|---|---|---|---|
| In-dist. | Out-of-dist. | 2,000 | 4,000 | Full-shot |
| CUB200 (224×224) | ImageNet | **59.68**±0.07 | **61.88**±0.04 | **65.56**±0.05 |
| | Places365 | 58.24±0.09 | 60.90±0.05 | 64.84±0.04 |
| CAR (32×32) | ImageNet | **65.82**±0.10 | **69.05**±0.07 | **72.14**±0.02 |
| | Places365 | 65.70±0.08 | 67.64±0.06 | 71.62±0.02 |