# OpenReview forum: "Task-Agnostic Undesirable Feature Deactivation Using Out-of-Distribution Data"
_NeurIPS.cc/2021/Conference — NeurIPS 2021 Poster_

### Official Review · Reviewer_MG8Y · 2021-07-09

**Rating:** 7
**Confidence:** 5

**Summary:**

Problem: DNN predictions may get influenced by the undesirable features that are irrelevant to the task at hand, which causes poor  generalization.
Solution: Existing work showed calibration on out-of-distribution (OOD) samples can remove the contribution of undesirable features in classification. The caveat is its applicability to feature extraction. The paper proposes TAUFE, a novel regularizer that deactivates all the undesirable features in OOD examples from the feature-extraction layer, thereby removing the dependency on the task specific softmax layer.
Results: TAUFE outperforms state-of-the-art methods on the standard benchmarks for classification, regression and the mix of the two.


**Ethical Concerns:**

None that are bothering at this point.

**Limitations And Societal Impact:**

Limitations are unknown performance for adversarial attacks.
In fact the authors should do a better job explaining the limitations and the societal impact.

**Main Review:**

Distinction from state-of-the-art:
The proposed approach clearly identifies the gap between the existing techniques to be applied for task agnostic feature extraction and the lack of flexibility and feature entanglement.

Pros:
The paper is well written, easy to understand and digest the concepts.

The very idea of task agnostic training of a deep model for feature extraction and then doing the tasks specific predictions is interesting.

The paper accurately highlights how deep models use easy features in images, video, object detection, etc, more often they can be irrelevant, in which case the proposed method can be more useful for accurate predictions.

Cons:

Since data augmentation seems to produce unbiased examples (as explained in [17]), how does TAUFE fare against those methods? Maybe why not compare against MixUp like [8] and show the effectiveness, at least for classification if not for regression?

Will TAUFE withstand the adversarial attacks? Maybe this is too much in a single paper, but it is worth throwing some light into the performance of TAUFE for adversarial attacks.


**Time Spent Reviewing:**

3 -- 4

---

> ### Author Response · Authors · 2021-08-10
> **Response to Reviewer MG8Y**
>
> We deeply appreciate the reviewers’ constructive comments and positive feedback on our manuscript.
>
> **Q1**. It would be possible to compare with semi-supervised methods based on data augmentation.
>
> This is a very good point. Since our main focus is to validate the effectiveness of TAUFE in the *standard* learning framework, we have chosen only basic supervised algorithms for the baselines. However, following the reviewer’s suggestion, we will conduct additional experiments on semi-supervised algorithms and add the results in the final version.
>
> **Q2**. Will TAUFE withstand the adversarial attacks?
>
> Thank you very much for helping us improve our paper. We expect that our regularizer is also effective in the *adversarial learning* setting as in OAT [8]. We will try to conduct additional experiments and include the results in the final version.

---

### Official Review · Reviewer_UyJi · 2021-07-12

**Rating:** 6
**Confidence:** 5

**Summary:**

This paper describes a simple yet effective approach to deactivate the undesirable features using out-of-distribution data. By shrinking the undesirable features from feature extraction layers instead of task-specific layers using L2 penalty, this method is task agnostic. Authors demonstrated and compared performance with task-specific approaches on classification, regression and mixed tasks using several benchmark data sets.

**Main Review:**

The key idea of this paper is to leverage L2 penalty to shrink the undesirable features extracted from OOD examples towards zero by assuming OOD and ID examples share such undesirable features. Compared with the recently published task-specific OAT approach, this method is task-agnostic yet simple to implement. Since L2 penalty shrinks the correlated features as groups, can authors provide some analysis to show the TAUFE approach de-activates groups of undesirable features but not much on the groups of desirable features? Authors provide Figure 2 illustration with some descriptions from lines 180-186. It would be great if authors can give in-depth analysis to show L2 penalty indeed shrinks undesirable feature groups without deactivating too much on desirable feature groups.

Related comment is on empirical analysis of the claimed properties using experiments. Authors successfully show TAUFE renders the activation of ID classes more distinguishable. However, it is necessary to empirically show that the undesirable feature groups have indeed been deactivated whereas not as much for the desirable feature groups. For example, authors might visualize the examples in desirable and undesirable feature spaces, respectively. Even if the method is demonstrated as task-agnostic analytically and empirically as requested, it can be OOD-dependent, i.e., the undesirable feature deactivation would depend on the choice of OOD examples. As such, the generalizability can be limited to the selected OOD examples from training only.

The reviewer is willing to raise the rating if authors successfully address the major concerns as mentioned above.

Minor comments:
Can authors also report experimental results for standard method using both ID and OOD examples?
TAUFE does not seem to outperform OAT across all settings. It is also not clear whether OAT has been appropriately optimized for its performance.

------------- New comments after reading authors' rebuttal
Authors have successfully addressed my concerns by providing an in-depth analysis to show the TAUFE approach deactivates the groups of undesirable features but not much on the groups of desirable features. As such, I raise my review rating to be positive.


**Time Spent Reviewing:**

4

---

> ### Author Response · Authors · 2021-08-10
> **Response to Reviewer UyJi**
>
> We deeply appreciate the reviewers’ valuable comments and some concerns. We hope that the concerns can be resolved through our clarifications in this rebuttal.
>
> **Q1**. Can authors provide some in-depth analysis to show the TAUFE approach deactivates the groups of undesirable features but not much on the groups of desirable features?
>
> Thank you very much for helping us improve our paper. Yes, we can provide a formal analysis for lines 164--186, as below. This in-depth analysis will be added in the supplemental material of the final version.
>
> Because a DNN extracts any type of features if it is statistically correlated with the target label $y$, a feature vector $f_{\phi}$ of an in-distribution example $x$ contains both a desirable feature $f_{desirable}$ and an undesirable feature $f_{undesirable}$.
> That is, $f_{\phi}(x)=f_{desirable}(x)+f_{undesirable}(x)$, where $f \in \mathbb{R}^d$.
> On the other hand, because an OOD example $\tilde{x}$ does not contain any features that semantically indicate the target label $y$, $f_{\phi}(\tilde{x})=f_{undesirable}(\tilde{x})$.
>
> For ease of exposition, let’s consider a binary classification setting. Let $x^+$ be an in-distribution example of class 1 and $x^-$ be an in-distribution example of class 2. Then, via standard learning, $f_{\phi}(x^+)=f_{desirable}(x^+)+f_{undesirable}(x^+)$ and $f_{\phi}(x^-)=f_{desirable}(x^-)+f_{undesirable}(x^-)$.
> Because $f_{undesirable}(x^+)$ and $f_{undesirable}(x^-)$ are expected to share some features with $f_{undesirable}(\tilde{x})$, both OAT and TAUFE attempt to reduce their effect by the regularization on $f_{\phi}(\tilde{x}) = f_{undesirable}(\tilde{x})$.
>
> For notational simplicity, let $f_{\phi}(x^+)=f_{desirable}^{+}+f_{undesirable}^+$ and $f_{\phi}(x^-)=f_{desirable}^{-}+f_{undesirable}^-$ (R3-1).
> Here, $f_{\phi}(x^+)$ and $f_{\phi}(x^-)$ correspond to the red circle and the blue circle, respectively, in Figure 2(a) of our paper.
>
> ***OAT***
>
> As analyzed in line 164--179 in our paper, OAT regularizes the undesirable features from OOD examples being activated into the decision boundary. Thus, each class feature in Eq. (R3-1) is forced to be changed as follows:
>
> $f_{\phi}^{OAT}(x^+)=f_{desirable}^{+}+(\alpha \frac{(f_{desirable}^{+}+f_{desirable}^-)}{2}  +f_{\perp}^+)$ and
> $f_{\phi}^{OAT}(x^-)=f_{desirable}^{-}+(\beta \frac{(f_{desirable}^{+}+f_{desirable}^-)}{2}  +f_{\perp}^-)$  (R3-2),
>
> where $\alpha, \beta \in \mathbb{R}$, $(f_{desirable}^{+}+f_{desirable}^-)/2$ is a vector on the decision boundary, and $f_{\perp}$ is an orthogonal vector to the plane basis of $f_{desirable}^+$ and $f_{desirable}^-$. Then,
>
> $f_{\phi}^{OAT}(x^+)=(1+\frac{\alpha}{2})f_{desirable}^++\frac{\alpha}{2}f_{desirable}^{-}+f_{\perp}^+$ and
> $f_{\phi}^{OAT}(x^-)=(1+\frac{\beta}{2})f_{desirable}^{+}+\frac{\beta}{2}f_{desirable}^{-}+f_{\perp}^-$      (R3-3).
>
> Therefore, because the undesirable feature $f_{undesirable}$ moves the activation of the desirable feature toward the decision boundary, these two types of the features (i.e., $f_{desirable}$ and $f_{undesirable}$) tend to be entangled, as illustrated in Figure 2(b).
>
> ***TAUFE***
>
> As analyzed in line 180--186 in our paper, TAUFE regularizes the undesirable features from OOD examples being deactivated on the feature space (i.e., toward the zero vector). Thus, each class feature in Eq. (R3-1) is forced to be changed as follows:
>
> $f_{\phi}^{TAUFE}(x^+)=f_{desirable}^{+}+\vec{0}$ and  $f_{\phi}^{TAUFE}(x^-)=f_{desirable}^{-}+\vec{0}$      (R3-4).
>
> Therefore, this regularization does not affect the activation of $f_{desirable}$, as illustrated in Figure 2(c), thereby encouraging a prediction of a DNN to be solely based on the desirable features. This concludes the theoretical analysis of our novel L2 penalty term.
>
> **Q2**. It is necessary to empirically show that the undesirable feature groups have indeed been deactivated whereas not as much for the desirable feature groups, e.g., by visualization.
>
> Referring to the visualization of Figure 2(a) in Andrew et al. (2019) where they disentangle the robust and non-robust features in adversarial learning, we have visualized the desirable and undesirable features learned by TAUFE. In our visualizations, the desirable features are easily comprehensible to humans for recognizing the original class (like the second row of Figure 2(a) in Andrew et al. (2019)), whereas the undesirable features are not (like the third row of Figure 2(a) in Andrew et al. (2019)). Unfortunately, there is no way to include the figures in the response. Instead, at our best effort, we show the pseudocodes used to get the desirable and undesirable features. We will certainly add those visualizations with an in-depth discussion in the final version.
>
> - - -
> Andrew, et al., 2019, Adversarial Examples Are Not Bugs, They Are Features, NeurIPS.
> - - -
> Algorithm R3-1: GetDesirableFeature(X)
>
> INPUT: $f_{\phi}^{TAUFE}$: a model learned by TAUFE, $X$: an in-distribution example, $\tilde{X}$: an OOD example, $N$: # of steps.
> OUTPUT: $X_d$: a reconstructed example of $X$ with only desirable features via $f_{\phi}^{TAUFE}$.
>
> 1: $X_d$ $\leftarrow$ $\tilde{X}$; /* initializing with $\tilde{X}$ */
>
> 2: **For** $i=1$ **to** $N$ **do**
> - $X_d$ $\leftarrow$ $X_d$ - $\alpha$ $\bigtriangledown$ $|| f_{\phi}^{TAUFE}(X_d)- f_{\phi}^{TAUFE}(X)||^2$
>
> 3: **return** $X_d$
> - - -
>
> Algorithm R3-2: GetUndesirableFeature(X)
>
> INPUT: $f_{\phi}$ and $f_{\phi}^{TAUFE}$: models learned by Standard learning and TAUFE, $X$: an in-distribution example, $\tilde{X}$: an OOD example, $N$: # of steps, $\epsilon$: a constant.
> OUTPUT: $X_u$: a reconstructed example of $X$ with only undesirable features via $f_{\phi}$ and $f_{\phi}^{TAUFE}$.
>
> 1: $X_d$ $\leftarrow$ $\tilde{X}$; /* initializing with $\tilde{X}$ */
>
> 2: **For** $i=1$ **to** $N$ **do**
> - $X_u$ $\leftarrow$ $X_u$ - $\alpha$ $\bigtriangledown$ $[ || f_{\phi}(X_u)- f_{\phi}(X)||^2 - max(\epsilon, || f_{\phi}^{TAUFE}(X_u)- f_{\phi}^{TAUFE}(X)||^2) ]$
>
> 3: **return** $X_u$
> - - -
>
> **Q3**.  Even if the method is demonstrated as task-agnostic analytically and empirically, it can be OOD-dependent, i.e., the undesirable feature deactivation would depend on the choice of OOD examples.
>
> You are absolutely right. In fact, we validated the effect of the choice of OOD datasets on TAUFE in the experiments. For example, we used 80mTiny and SVHN as OOD datasets for CIFARs, and ImageNet-990 and Places365 for ImageNet-10. Overall, we tested *eight* different combinations of in-distribution and OOD datasets for classification (see Table 2) and *four* different combinations for regression (see Table 3) and for a mix of classification and regression (see Table 4). We will further clarify the presentation in the final version.
>
> **Q4**. Can authors also report experimental results for the standard method using both ID and OOD examples?
>
> Thank you for your interesting question. Since OOD examples do not have any target label, the standard learning with only Cross-Entropy (CE) loss cannot directly use OOD examples for optimization. Thus, to answer the reviewer’s question, we assign all OOD examples to a meta-class. That is, the classifier is trained to return the (K+1)-th class for OOD examples, where K is the number of in-distribution classes. We have observed that the classification accuracy of this variant degrades to 72.37 for CIFAR10 under the full-shot setting. If a test example contains a lot of undesirable features, it can be incorrectly predicted as the meta-class. Overall, a straightforward method of using OOD examples may even harm the classification accuracy. We will add this analysis result in the final version.
>
> **Q5**.  Is OAT appropriately optimized for its performance?
>
> Yes, we believe that OAT is appropriately optimized. We followed its optimal setting suggested in the original paper [8]. Moreover, we have performed cross-validation for the experiments and observed that the accuracy improves by around 1 pp in all tests, as shown in Table R3-1. The accuracy values in the original paper [8] are reproduced in our experiments. Of course, TAUFE consistently outperforms OAT with cross-validation. We will accordingly update the results in the final version.
>
> | Dataset | Method | | In OAT [8] | w. cross-validation |w/o. cross-validation |
> |:--------------:|:-------:|---|:-------:|:-------:|:-------:|
> |CIFAR10/80mTiny|Standard| |94.46|94.22|93.84|
> |                             |OAT| |95.20|95.11|93.95|
> |                             |TAUFE| |-|95.57|94.01|
> |CIFAR100/80mTiny|Standard| |73.87|73.84|73.40|
> |                             |OAT| |76.30|76.27|75.81|
> |                             |TAUFE| |-|76.53|76.03|
>
> Table R3-1: Validation of the OAT performance in our experiments.

---

### Official Review · Reviewer_d4nE · 2021-07-13

**Rating:** 6
**Confidence:** 4

**Summary:**

The paper proposes a simple regularization that encourages logits to be close to zero on a designated out-distribution during training. The authors show that this improves accuracy across various tasks and datasets.

**Ethical Concerns:**

There is the issue of using 80mTiny as an out-distribution since the dataset has been retracted by the authors over ethical concerns. Arguably, it is okay to use it for comparison with old work if the authos also provide results on a different OOD dataset that can be used for future works to compare with. Fortunately, the authors do provide experiments using SVHN as well. However, this issue needs to be discussed at the very least. As it stands right now I cannot agree with the author's sentiment in point 1. d) of the checklist that their paper conforms to the ethics guidelines.

**Limitations And Societal Impact:**

I am not quite sure why the authors chose to move their limitations section into the appendix. They clearly still had plenty of space on the final page and the page limit was extended to 9 pages in part to allow for this sort of discussion in the main paper. I recommend moving this back into the main paper.

**Main Review:**

Strengths:
The proposed method is quite simple and the results in the paper indicate that it can be very effective at boosting accuracy. The method is also task-agnostic which means it can be applied beyond just, e.g. image classification. Also, the authors have used several different training out-distributions which gives more information about the dependence on the choice of this distribution than many other papers in the field. The inclusion of few-shot results is also interesting.


Weaknesses:
A major issue with this paper is the weakness of their baseline. Even when only using a ResNet18 on CIFAR10 a plain trained classifier can attain 95.0% using AutoAugment. Similarly, an outlier exposure (OE being, as far as I can tell, identical to the OAT that they compare to) trained model can achieve 95.5% when using the same 80mTiny dataset as out-distribution as the authors. On CIFAR100 (77.4% and 77.3%, respectively). In other words, their improvement beyond their baseline is much smaller than their gap to a simple model trained using AutoAugment. Would their results continue to hold when using stronger baselines?

Another issue is that they only compare the classification performance in terms of accuracy. Outlier-aware methods are typically designed to improve out-of-distribution detection. Does their method perform better or worse than OE/OAT on this task? Even if it is not the main focus of the paper, these results need to at least be discussed and included in the appendix.

Since the focus of the paper is so heavily on improving the performance on the original task, I also think that they need to include baselines from the semi-supervised literature (for example, [1], but hopefully also more recent works).


Clarity:
The clarity of the paper needs much improvement. Especially experimental details are not fully laid out, e.g. line 236 "best values are obtained via a grid search." What was the grid? How were the validation sets chosen? What was the performance metric that they optimized for?

I also find Table 1 a little unclear. The caption states "Average cosine similarity between each classe’s activation on CIFAR-10 for Standard, OAT,and TAUFE". I understand this to mean that the cosine similarity between the features of samples within the same class was computed and that those results were averaged across classes. If I understand this correctly, then I do not understand how this is a bad thing. Higher similarity within classes does not seem like an issue. The accompanying discussion in the main text seems to indicate that the authors are instead talking about the cosine similarities between the different classes. How is this computed exactly? The mean of each class is computed and those are compared? Or the cosine similarities with samples from other classes are computed and then averaged?

The authors also don't explain what ImageNet-10 is. They write in line 222 that it consists of "randomly selected classes". Which classes? This needs to be in the appendix.


Reproducibility:
I very much appreciate that the authors provided their code in the submission.


Additional feedback:
The definitions of desirable and undesirable features should probably be modified slightly to be called rho-desirable or rho-undesirable. This is especially important in Eq. 2 where F_out is defined in terms of a rho, that is not explicitly given. The authors clearly meant to assume that rho is the same for both definitions but the way it is written now is sloppy. Also, the discussion below it should maybe point out, that "a feature vector [...] could be a mixture of desirable and undesirable features" but that it could very well contain neither.

Another point that I would appreciate some clarification on is in line 240 where the authors state that they only sample N points from the out-distribution - even in the full-shot regime. This appears somewhat non-standard so I would have liked some justification for this choice in the experimental setup.

[1] There are many consistent explanations of unlabeled data: Why you should average; Ben Athiwaratkun, Marc Finzi, Pavel Izmailov, Andrew Gordon Wilson; at ICLR'19

**Time Spent Reviewing:**

4

---

> ### Author Response · Authors · 2021-08-10
> **Response to Reviewer d4nE**
>
> We deeply appreciate the reviewers’ valuable comments and reasonable concerns. We hope that they can be resolved through our clarifications in this rebuttal.
>
> **Q1**. A major issue is the weakness of the baselines. Would the results continue to hold when using stronger baselines?
>
> We appreciate the reviewer for pointing out this issue. This work aims at developing a novel regularizer for OOD examples. The baseline, OAT [8], is the state-of-the-art algorithm for this direction, as far as we know. That is, we believe that OAT is not a weak baseline. Moreover, the performance numbers of AutoAugment are not directly comparable to our performance numbers, because AutoAugment performs sophisticated data augmentation whereas TAUFE does not perform any kind of data augmentation. Thus, we expect that the absolute performance values will increase when data augmentation and cross-validation are additionally performed. As quick evidence, we have observed that the classification accuracy of TAUFE (CIFAR10/80mTiny) increases from 94.01 to 95.57 simply by performing cross-validation.
>
> **Q2**. Does TAUFE perform better or worse than OE/OAT on the OOD detection task?
>
> Thank you very much for your insightful comment. Even if the OOD detection performance is not the main focus of this paper, we have measured the OOD detection performance before and after the proposed regularizer of TAUFE is incorporated into two representative OOD detection methods: uncertainty-based [Hendrycks et al., 2019] and energy-based [Liu et al., 2020]. As shown in Table R2-1, the OOD detection performance is not that affected by TAUFE. We will add these results in the supplemental material of the final version.
>
> |        |OOD score|   ||      Uncerainty        ||  |          |Energy|          |
> |:-----------:|:-------:|---|:-------:|:-------:|:-------:|---|:-------:|:-------:|:-------:|
> |**Dataset**    |**Method**   |  |**AUROC**|**AUPR_out**|**FPR95**| |**AUROC**|**AUPR_out**|**FPR95**|
> |CIFAR10  |Standard|  |92.20|88.56|20.67|  |93.6|89.96|20.06|
> |                 |TAUFE   |  |92.17|89.71|22.96|  |93.37|90.08|22.88|
> |CIFAR100|Standard|  |83.31|79.37|45.76|  |88.46|86.04|35.35|
> |                 |TAUFE   |  |82.03|79.21|47.89|  |88.42|85.56|37.82|
>
> Table R2-1: OOD detection performance of TAUFE compared with representative OOD detection methods. We use 80mTiny as the OOD dataset which is exposed in the training phase, and use SVHN for measuring the detection performance.
>
> - - -
> Hendrycks, et al., 2019, Deep Anomaly Detection with Outlier Exposure, ICLR.
> Liu, et al., 2019, Energy-based Out-of-distribution Detection, NeurIPS.
> - - -
>
> **Q3**. The authors need to include more baselines from semi-supervised literature.
>
> This is a very good point. Since our main focus is to validate the effectiveness of TAUFE in the *standard* learning framework, we have chosen only basic supervised algorithms for the baselines. However, following the reviewer’s suggestion, we will conduct additional experiments on semi-supervised algorithms and add the results in the final version.
>
> **Q4**. The clarity of the paper needs much improvement.
>
> Thank you for carefully checking the clarity of the paper. We will clarify the meaning of cosine similarity in Table 1. Simply speaking, it was the average of the cosine similarities between all pairs across different classes. In addition, we will clarify the ten classes sampled for ImageNet-10. These issues can be resolved very easily.
>
> Furthermore, we will significantly enrich the supplemental material by adding more details on hyper-parameter tuning, including the grid, validation set, and performance metrics. Besides, we will add the results of the sensitivity analysis for different hyper-parameter values.
>
> **Q5**. The definitions of desirable and undesirable features should probably be modified slightly.
>
> Thank you again for your careful comment. We intended $\rho$ in Eq. (1) and $\rho$ in Eq. (2) to be the same, but your point makes more sense for rigorous definitions. We will fix them following your suggestion in the final version.
>
> **Q6**. Only sampling N points from OOD even in the full-shot regime needs some justification.
>
> Thank you for your comment. There is no standard practice for the amount of OOD examples, and we simply follow the setting used in OAT [8] for fair comparison. However, it is possible to add more OOD examples than in-distribution examples, and we will explore this setting as future work.
>
> **Q7**. There is an ethical issue of using 80mTiny dataset.
>
> We overlooked this ethical issue on the 80mTiny dataset. We will definitely discuss this issue in the final version so that readers and future researchers can consider it.

---

> > ### Author Response · Authors · 2021-08-20
> > **Additional Response to Reviewer d4nE**
> >
> > We again deeply appreciate the reviewers' careful comments.  To further facilitate the discussion phase, we provide the full experiment result regarding the reviewer d4nE's concern on a weak baseline (**Q1**).
> >
> > **Q1**. A major issue is the weakness of the baselines. Would the results continue to hold when using stronger baselines?
> >
> > We have quickly strengthened the baseline model by performing cross-validation to achieve better optimization.  Specifically, we conducted 20-fold cross-validation and selected the model with the best validation accuracy for testing.  As a result, it happens that the accuracy improves by around 1 pp in all tests, as shown in Table R2-2.  Of course, **TAUFE consistently outperforms OAT**.  Thus, we believe that the result will continue to hold for stronger baselines (e.g., with data augmentation).
> >
> > | Dataset | Method | | w. cross-validation (improved baseline) |w/o. cross-validation (original baseline) |
> > |:--------------:|:-------:|---|:-------:|:-------:|
> > |CIFAR10/80mTiny|Standard| |94.22|93.84|
> > |                             |OAT| |95.11|93.95|
> > |                             |**TAUFE**| |**95.57**|94.01|
> > |CIFAR100/80mTiny|Standard| |73.84|73.40|
> > |                             |OAT| |76.27|75.81|
> > |                             |**TAUFE**| |**76.53**|76.03|
> >
> > Table R2-2: Performance comparison on top of an improved baseline model.

---

> > > ### Comment · Reviewer_d4nE · 2021-09-10
> > > **Raised Score**
> > >
> > > I appreciate the authors response, clarification and additional experiments.
> > > I raise my score and agree to acceptance under the condition that the authors include the SSL baselines, as promised.

---

### Official Review · Reviewer_32rF · 2021-07-19

**Rating:** 7
**Confidence:** 4

**Summary:**

TAUFE, a novel regularizer has been presented to handle OOD. It rigorously validates its performance on three tasks, classification, regression, and a mix of them, on CIFAR-10, CIFAR-100, ImageNet, CUB200, and CAR datasets with promising results.

**Ethical Concerns:**

Plagiarism is 10% as seen in Docoloc - https://www.docoloc.de/d88c4d5162dd4b21eeb6add446e2b5bbCEuFHG85rulAsphkx9/en/konto.hhtml?dogetresult=52 - hence OK - cleared - green signal.

**Limitations And Societal Impact:**

Not applicable here.

**Main Review:**

Novelty:
The paper overall is a novel take on OOD problem. TAUFE, a novel regularizer claims to deactivate undesirable features using OOD examples in the feature extraction layer and thus removes the dependency on the task-specific softmax layer.

Strengths:
1. Code is given zipped as well as URL for verification.
2. Apart from classification tasks, where OOD has started showing some grip, this work also takes care of regression and a mix of them.
3. The experiments and validation on standard datasets creates a good impression wrt results.
4. Mathematical formulation and comparison with softmax layer is sound.

Weakness:
1. there was ample scope to add finer details of implementation and ablation studies in the supplementary section.
2. Ablation on self to understand the pitfalls where method is failing and identifying the corner and failure cases will have made the paper better as well as laid path for future work in need to fill gaps.
3. "deactivates all undesirable features " - is this term not too bold to say?

Relevance to NeurIPS:
Very relevant, aligned to hot area of research "out-of-distribution" handling.

Clarity:
The paper is well written.

Related work Comparison:
There is substantial comparison with other works and standard datasets.

Effort: In terms of thought process, experiments effort is there. However some effort on analysis and corner case / failure identification would have made this a very good effort.

Suggestions:
1. Probably, varying the types of data of the datasets (datatype) might help in getting deeper in the problem as a generalization tool.]
2. Some explanation needed what is meant by undesirable features and desirable features? Can a bias on desirable features help in drawing the boundary line? Is there a possibility of a membership function based boundary line?
3. How will these work in adverserial examples?


**Time Spent Reviewing:**

3

---

> ### Author Response · Authors · 2021-08-10
> **Response to Reviewer 32rF**
>
> We deeply appreciate the reviewers’ constructive comments and positive feedback on our manuscript.
>
> **Q1**. There was ample scope to add finer details of implementation and ablation studies.
>
> Thank you very much for your comment. Yes, we will significantly enrich the supplemental material by adding more details on hyper-parameter tuning, including the grid, validation set, and performance metrics, as well as algorithm implementation. Besides, we will add the results of the sensitivity analysis of the hyper-parameter $\lambda$, which is the only hyper-parameter specific to TAUFE, as shown in Table R1-1. The value 0.1 used for all experiments is shown to achieve the best accuracy.
>
> |$\lambda$|0.01|**0.1**|1|10|
> |:-----------:|:-------:|:-------:|:-------:|:-------:|
> |CIFAR10/80mTiny|74.22|**74.42**|74.17|73.64|
> |CIFAR100/80mTiny|25.30|**25.50**|25.23|24.83|
>
> Table R1-1: The effect of $\lambda$ on TAUFE’s performance for CIFARs under 2,500-shot settings.
>
> **Q2**. Ablation on self to understand the pitfalls where the method is failing and identifying the corner and failure cases will have made the paper better as well as laid a path for future work in need to fill gaps.
>
> Thank you very much for helping us improve our paper. As you suggested, we will try to consider a wider range of datasets to find potential corner cases. Along the same line, we will also investigate the effect of TAUFE under the adversarial learning setting. These potential corner cases will be discussed in the supplemental material of the final version.
>
> **Q3**. "deactivates all undesirable features" - is this term not too bold to say?
>
> We agree with you and will rephrase “all” with “many.”
>
> **Q4**. Probably, varying the types of data might help in getting deeper in the problem as a generalization tool.
>
> The undesirable features can exist in other types of data as we discussed in the related work section. This suggestion would be a great future research direction to help in getting deeper in the problem, and we will discuss this issue in the final version.
>
> **Q5**. Some explanations are needed. What is meant by undesirable features and desirable features? Can a bias on desirable features help in drawing the boundary line? Is there a possibility of a membership function based boundary line?
>
> Thank you very much for pointing out these unclear points. We will definitely improve the presentation of the final version to address these questions. In this author's feedback, let us quickly answer these questions.
> - An undesirable feature is a pattern that is somehow statistically correlated with the target label, but meaningless in semantics (e.g., ‘desert’ background for recognizing a ‘camel’). In contrast, a desirable feature is a pattern that is indeed meaningful to indicate the label (e.g., ‘hump’ shape for recognizing a ‘camel’). These concepts are formalized by Definition 3.1 and Definition 3.2.
> - Yes, as shown in our experiments (see Figure 3(c)), the boundary drawn by desirable features shows better generalization on test data.
> - Yes, since we use the softmax scores to determine membership labels, we can regard that our classifier produces membership function-based boundary lines.
>
> **Q6**. How will these work in adversarial examples?
>
> We expect that our regularizer is also effective in the adversarial learning setting as in OAT [8]. We will try to conduct additional experiments and include the results in the final version.

---

### Review · Ethics_Reviewer_1QET · 2021-08-31

**Recommendation:** My recommendation is contained above.

**Ethical Issues:**

Yes

**Ethics Review:**

As noted by reviewer d4nE this paper uses the 80mTiny dataset which has been retracted due to concerns about 'racist, sexist, and otherwise offensive labels' (see: https://venturebeat.com/2020/07/01/mit-takes-down-80-million-tiny-images-data-set-due-to-racist-and-offensive-content/). As such, it represent the use of a dataset that has been discredited by its creators (see the ethics guidelines for a caution against this practice). Typically, NeurIPS submissions that are based upon discredited data-sets are asked to remove these datasets from the paper and to substitute them with ones that have been responsibly sourced.

If it is not possible to perform this action, then the authors must state their explicit rationale for using the dataset, and include warnings in a social impact statement. Given that it is an important part of the paper, my recommendation is that this discussion take place in the text itself, rather than in the supplementary materials, as failure to explicitly acknowledge the risk could lead to further propagation/use of the discredited dataset. There should be a recommendation that future work focuses only on SVHN.

I will leave it to the other reviewers to determine whether the latter option is sufficient to warrant approval in this case.

---

> ### Author Response · Authors · 2021-09-01
> **Response to Ethics Review**
>
> Thank you very much for carefully checking the ethical issue. **We will make sure to remove the 80mTiny dataset and substitute it with another dataset** (e.g., LSUN [Yu et al., 2015]). Because the SVHN dataset was also used for the tests associated with the 80mTiny dataset, the superiority of the proposed algorithm TAUFE over the state-of-the-art algorithm OAT has been shown even without the 80mTiny dataset. We used the 80mTiny dataset simply because OAT (published in ICLR 2021) had used it.
>
> Yu et al., 2015, LSUN: Construction of a Large-Scale Image Dataset using Deep Learning with Humans in the Loop, arXiv:1506.03365.

---

### Decision · Program_Chairs · 2021-09-27

**Decision:**

Accept (Poster)

**Comment:**

This paper proposes a simple regularisation that encourages features to be close to zero on a pre-specified OOD training set.

Reviewers think the proposed approach is novel, simple, and effectively demonstrated on a variety of experiments. An ethical issue has been raised on the 80mTiny dataset used in experiments, for which the authors promise to remove it. So it comes down to the question whether the remaining result is significant enough to demonstrate the claims, for which the reviewers do not see it as an issue.

Because of this issue with the 80mTiny dataset, the paper is being marked as conditionally accept.

Reviewers raise another concern on missing comparison to semi-supervised learning approaches, and the authors agree to add in experiments in revision. Label/logit smoothing and feature denoising have also been extensively discussed in adversarial robustness literature, it would improve the paper if the authors can evaluate the proposed approach on adversarial examples.

Also I would suggest to openly discuss the limitation of the proposed approach in the main text rather than in appendix.

----

UPDATE: Upon reviewing the revision, the decision has been updated to accept.